# Neutron Lifetime Anomaly and Mirror Matter Theory

Wanpeng Tan 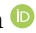

Department of Physics and Astronomy and Institute for Structure and Nuclear Astrophysics (ISNAP), University of Notre Dame, Notre Dame, IN 46556, USA; wtan@nd.edu

**Abstract:** This paper reviews the puzzles in modern neutron lifetime measurements and related unitarity issues in the CKM matrix. It is not a comprehensive and unbiased compilation of all historic data and studies, but rather a focus on compelling evidence leading to new physics. In particular, the largely overlooked nuances of different techniques applied in material and magnetic trap experiments are clarified. Further detailed analysis shows that the "beam" approach of neutron lifetime measurements is likely to give the "true" $\beta$-decay lifetime, while discrepancies in "bottle" measurements indicate new physics at play. The most feasible solution to these puzzles is a newly proposed ordinary-mirror neutron ($n - n'$) oscillation model under the framework of mirror matter theory. This phenomenological model is reviewed and introduced, and its explanations of the neutron lifetime anomaly and possible non-unitarity of the CKM matrix are presented. Most importantly, various new experimental proposals, especially lifetime measurements with small/narrow magnetic traps or under super-strong magnetic fields, are discussed in order to test the surprisingly large anomalous signals that are uniquely predicted by this new $n - n'$ oscillation model.

**Keywords:** mirror matter theory; mirror symmetry; spontaneous symmetry breaking; neutron lifetime; CKM unitarity; particle oscillations; topological transitions; laboratory tests; beyond the Standard Model

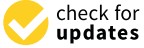



## 1. Introduction

The neutron lifetime puzzle has been an evolving and complicated issue since high precision lifetime measurements of $\lesssim 3$ s in uncertainties began in the late 1980s (see the reviews in Refs. [1,2] and a historic note from Ref. [3]). Considering that the sole known decay mode of neutrons is $\beta$ decay, one would have thought that tremendous technological advances over the years would make it easy to measure the neutron lifetime. However, these modern results with uncertainties of about 10 s or less since the late 1980s are full of inconsistencies as demonstrated in Figure 1 and Table 1, which seem to be the ecstatic signs of new physics.

### 1.1. 1% Discrepancy in Neutron Lifetime Measurements

Probably the most notable anomaly, especially in recent years, is the about 1% discrepancy between two sets of high precision measurements via the so-called "beam" and "bottle" approaches, respectively. Specifically, the BL1 experiment [4,5], using the "beam" approach, measures the neutron flux directly from a cold neutron beam while detecting the emitted protons from $\beta$ decays—the only known decay mode of neutrons. It measures the $\beta$ decay rate directly, as long as other hidden neutron-disappearing processes are less significant than its precision level (roughly, $10^{-3}$ or below). This approach typically and consistently gives a neutron lifetime of about 888 s and its current average value of $888.2 \pm 2.0$ s is depicted by the orange shaded band in Figure 1.

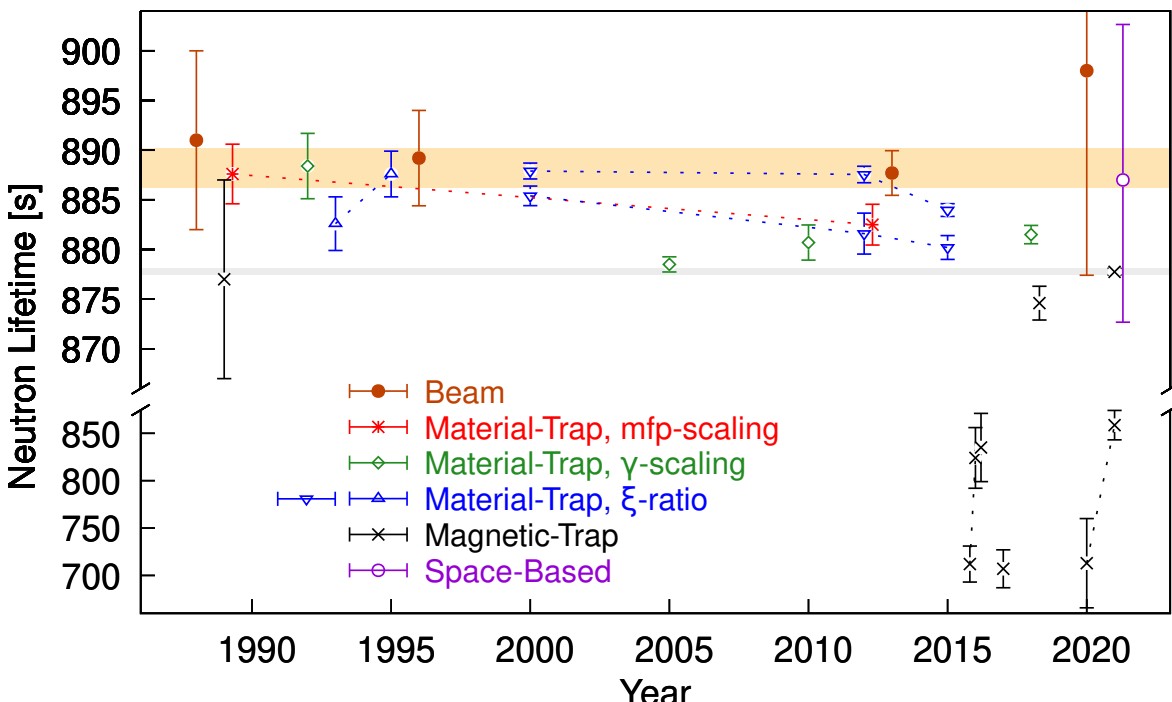

**Figure 1.** Results of modern neutron lifetime measurements are shown. Different techniques are presented with different symbols and colors. Correlated results from the same apparatus are connected by dashed lines. See detailed information in text and Table 1.

On the other hand, the "bottle" experiments store ultra-cold neutrons (UCN) confined by gravitational force or magnetic fields in a material or magnetic trap [6–8]. By measuring the neutron loss rate in the trap, this method typically presents a neutron lifetime of about 880 s. Note that any other unknown loss processes in the trap will contribute to the measured lifetime and make it appear shorter. The 1% difference between the results of the two approaches has become more severe (at a $4.4\sigma$ level) recently with the most precise "beam" measurement of $887.7 \pm 1.2(\text{stat}) \pm 1.9(\text{sys})$ s at NIST [5] and the newly updated magnetic trap value of $877.75 \pm 0.28(\text{stat}) + 0.22/-0.16(\text{sys})$ s by UCN$\tau$ collaboration [9].

However, a more popular trend in the community is to try to dismiss or ignore the lifetime anomaly. Such voices seem to be increasingly prevalent in favor of the "bottle" approach, despite a lack of clear and forceful evidence to support it. For example, the Particle Data Group (PDG) has excluded many of the seemingly "outlandish" results from the calculation of their recommended lifetime value since 2019 [10]. These discredited data include all the "beam" approach measurements and other results that support the averaged "beam" value. Most recently, systematic corrections applied in the best "beam" measurements using the BL1 apparatus at NIST [5] were questioned by Ref. [11], in particular, mechanisms involving possible proton losses. However, these criticisms were firmly rejected by the NIST BL1 collaboration shortly [12]. The 1% anomaly seems to persist and should be taken more seriously. As discussed below, the issues, or better yet new physics, are most likely present in the "bottle" measurements.

### 1.2. More Puzzles in Neutron Lifetime and CKM Unitarity

In fact, the neutron lifetime puzzle is more complicated than the 1% discrepancy between the two approaches mentioned above. A long-standing anomalous loss effect from solid walls of a material UCN trap has been observed and has puzzled the UCN community for many decades [3,13]. According to quantum mechanics, the total reflection of UCNs from walls, which are made of good clean trapping materials such as beryllium, should be ideal, at least at low enough temperatures, for storage of UCNs. However, the large unexpected loss rate is observed at a loss level of about $10^{-5}$ per reflection and temperature

independent [14]. The loss mechanism is still unknown, but inelastic up-scattering or quasi-elastic heating, and other proposed mechanisms seem to be ruled out by both theoretical calculations and experimental measurements as these rates are several orders of magnitude lower than the anomalous yet persistent loss rate [15–17].

In addition to the disagreement with the "beam" approach, there have been inconsistent results even within the same "bottle" approach. For example, the first high-precision "bottle" measurement published by Mampe et al. in 1989 reported a lifetime value of $887.6 \pm 3.0$ s, which included an even smaller statistical error of 1.1 s. This value is at odds with most "bottle" results but is almost identical to the current "beam" value. Even the two most recent high-precision "bottle" measurements, giving lifetime values of $877.75 \pm 0.28(\text{stat}) + 0.22/-0.16(\text{sys})$ s from UCN$\tau$ collaboration [9] and $881.5 \pm 0.7(\text{stat}) \pm 0.6(\text{sys})$ s with Gravitrap II [7], respectively, exhibit a strong tension at the $3.8\sigma$ level.

Due to the anomalous loss effect from walls mentioned above, results using material traps have to be extrapolated. The several different extrapolation techniques seem to produce, at times, discrepant results because of their differences in model dependence, possibly indicating the existence of new physics. Sometimes, such discrepancies are at a significance level even beyond $5\sigma$ (e.g., between data of Arzumanov2000 [18] and Serebrov2005 [19] as shown in Table 1). More details of such inconsistencies will be elaborated on in the next section.

Ideally, magnetic traps are free of the loss problem from trap walls and should have provided more consistent results if no new physics is involved. In reality, all the small or narrow magnetic traps have obtained much lower lifetime values, sometimes more than 100 s lower, and unfortunately, some of them were even discontinued, possibly because of too anomalous results. For example, the pioneering Ioffe-type magnetic trap at NIST produced an astonishing final result before it was discontinued, an anomalous lifetime value of $707 \pm 20$ s reported in an unpublished thesis work [20]. The only successful one (in terms of producing high-precision results consistent with mainstream values) is the UCN$\tau$ project, which is a large device. Is new physics dependent on trap geometry? More working magnetic traps will definitely help address this question.

The neutron lifetime puzzle is also closely related to the CKM unitarity problem. One of the major reasons why the "bottle" values are favored is perhaps a misunderstanding of the CKM unitarity. There are many different approaches to measuring the matrix elements of CKM. Roughly, it can be categorized into three different systems: mesons, neutrons, and nuclei. Very often, results from different systems are mixed together, making things unclear. Reanalysis by carefully separating different results from different systems [21] has clearly shown where the issues are, i.e., the "bottle" lifetime value agrees better with nuclear $0^+ \to 0^+$ data while the "beam" lifetime supports meson data. One can imagine that meson data could be cleaner or free from possible new physics effects in such a simple particle system while it may be the opposite in a more complicated nuclear system. At least, solving the neutron lifetime puzzle will also settle down the CKM unitarity issue or vice versa.

**Table 1.** Results of modern neutron lifetime measurements are compiled and categorized according to their applied methods. Quoted lifetime values include the following types: $\tau_n$—measured lifetime; $\tau_\beta$—beta decay lifetime for the "beam" approach; $\tau_{un}$—uncorrected lifetime; $\tau_s$—storage lifetime (without systematic corrections applied) for some magnetic trap measurements.

| Approach | Sub-Method | Device/Location | Ref. | Lifetime Results [s] |
|---|---|---|---|---|
| beam | p-extraction | KIAE | Spivak1988 [22] | $\tau_\beta = 891 \pm 9$ |
| | quasi-Penning trap | ILL<br>BL1/NIST | Byrne1996 [23]<br>Yue2013 [5] | $\tau_\beta = 889.1 \pm 4.8$<br>$\tau_\beta = 887.7 \pm 1.2_{stat} \pm 1.9_{sys}$ |
| | TPC/pulsed | J-PARC | Hirota2020 [24] | $\tau_\beta = 898 \pm 10_{stat}(^{+15}_{-18})_{sys}$ |
| material trap | mfp scaling | MAMBO/ILL | Mampe1989 [25]<br>Steyerl2012 [26][1] | $\tau_n = 887.6 \pm 3.0$<br>$\tau_n = 882.5 \pm 1.4_{stat} \pm 1.5_{sys}$ |
| | $\gamma$-scaling | PNPI<br>Gravitrap/ILL<br>MAMBO II/ILL<br>Gravitrap2/ILL | Nesvizhevskii1992 [27][2]<br>Serebrov2005 [19]<br>Pichlmaier2010 [28]<br>Serebrov2018 [7] | $\tau_n = 888.4 \pm 3.1_{stat} \pm 1.1_{sys}$<br>$\tau_n = 878.5 \pm 0.7_{stat} \pm 0.3_{sys}$<br>$\tau_n = 880.7 \pm 1.3_{stat} \pm 1.2_{sys}$<br>$\tau_n = 881.5 \pm 0.7_{stat} \pm 0.6_{sys}$ |
| | $\xi$-ratio | ILL | Mampe1993 [29]<br>Ignatovich1995 [30][3] | $\tau_n = 882.6 \pm 2.7$<br>$\tau_n = 887.6 \pm 2.3$ |
| | | KIAE-double<br>bottle/ILL | Arzumanov2000 [18] | $\tau_{un} = 887.9 \pm 0.8$<br>$\tau_n = 885.4 \pm 0.9_{stat} \pm 0.4_{sys}$ |
| | | | Arzumanov2012 [32] | $\tau_{un} = 887.55 \pm 0.83$<br>$\tau_n = 881.6 \pm 0.8_{stat} \pm 1.9_{sys}$ |
| | | | Arzumanov2015 [8] | $\tau_{un} = 883.97 \pm 0.64$<br>$\tau_n = 880.2 \pm 1.2$ |
| magnetic trap | storage ring | NESTOR/ILL | Paul1989 [33] | $\tau_n = 877 \pm 10$ |
| | Halbach | HOPE/ILL | Leung2016 [34] | $\tau_s(\text{no remover}) = 712 \pm 19$<br>$\tau_s(80\text{ cm}) = 824 \pm 32$<br>$\tau_s(65\text{ cm}) = 835 \pm 36$<br>$\tau_n(\text{wall losses}) = 887 \pm 39$ |
| | | PNPI/ILL | Ezhov2018 [35] | $\tau_s = 874.6 \pm 1.7$ |
| | | $\tau$SPECT/Mainz | Kahlenberg2020 [36]<br>Ross2021 [37] | $\tau_s = 713 \pm 47$<br>$\tau_s = 859 \pm 16$ |
| | | UCN$\tau$/LANL | UCN$\tau$2021 [9] | $\tau_n = 877.75 \pm 0.28_{stat}(^{+0.22}_{-0.16})_{sys}$ |
| | Ioffe | NIST | Huffer2017 [20] | $\tau_n = 707 \pm 20$ |
| space | Moon-based | LPNS | Wilson2021 [38] | $\tau_n = 887 \pm 14_{stat}(^{+7}_{-3})_{sys}$ |

### 1.3. Possible Theoretical Solutions and New Experimental Efforts

To solve the neutron lifetime anomaly, a number of ideas involving exotic particle oscillations or dark decays have been pursued as far as new physics is concerned. The idea of $n - \bar{n}$ oscillations, though intriguing from the point of view of beyond-the-Standard-Model physics, is unlikely to settle the issue due to the well-constrained oscillation time scale of $\tau_{n\bar{n}} > 0.86 \times 10^8$ s in an early experiment [39]. A recent attempt to consider neutrons that decay to particles in the dark sector showed an interesting decay channel of $n \to \chi + \gamma$ with constraints of 937.900 MeV $< m_\chi <$ 938.783 MeV for the dark particle mass and 0.782 MeV $< E_\gamma <$ 1.664 MeV for the photon energy [40]. Unfortunately, such a possibility was dismissed shortly by an experiment [41] and a similar channel of $n \to \chi + e^+ + e^-$ was excluded as well [42].

By introducing a six-quark coupling in the mirror matter theory for the $n$ and $n'$ interaction of $\delta m \sim 10^{-15}$ eV with a large mass cutoff at $M \sim 10$ TeV, Berezhiani and Bento proposed a possible $n - n'$ oscillation mechanism with a time scale of $\tau \sim 1$ s [43]. Later on, such oscillations were refuted experimentally with a much higher constraint of $\tau \geq 448$ s [44–47]. A revised $n - n'$ oscillation model was proposed using both the six-quark coupling and a small $n - n'$ mass splitting of $10^{-7}$ eV [48] where, like many other studies, the "bottle" lifetime is favored. Not surprisingly, again, it was basically ruled out by a null result from a recent publication in searching such oscillations under magnetic fields [49].

An immediate puzzle to be resolved is determining which of the two different types of measurements provides the true beta decay lifetime for neutrons. The mainstream support goes to the "bottle" approach. Part of the reason is related to the CKM unitarity issue that will be discussed in detail in Section 2.5. As we shall see, this may be one of the biases to overcome in order to discover new physics.

Different models aim at different ways to solve the above puzzles. As we will elaborate in Section 3, a more promising solution could be the newly proposed $n - n'$ oscillation model [21,50], which regards the "beam" lifetime as the true $\beta$-decay lifetime. In this rather exact new mirror matter model of two parameters, no cross-sector interaction is introduced, unlike other models, and spontaneous mirror symmetry breaking is the sole reason responsible for ordinary-mirror particle oscillations. The new model and further developed framework of the underlying mirror matter theory can consistently and quantitatively explain many puzzles and various observations in fundamental physics, astrophysics, and cosmology [21,50–61]. Most intriguingly, various feasible laboratory tests, including new lifetime measurements, have been proposed to verify the unique predictions of the new model [21,58].

On the experimental front, much effort is focused on a better understanding of experimental systematics and proposals for device upgrades and new methods for neutron lifetime measurements. In particular, the "beam" approach will be improved in the near future. For example, the BL2 project developed for better in-beam lifetime measurements at NIST will improve upon its predecessor, BL1, and, hopefully, reduce uncertainties to less than one second [62]. Furthermore, in the future, BL3, a planned upgrade as a scaled-up device of BL2, has goals of further reducing errors to $\lesssim 0.3$ s [63].

A small magnetic trap $\tau$SPECT [36,37] has just begun to produce preliminary results, and there are also efforts to revive other small magnetic traps. Another large magnetic trap, PENeLOPE [64], with a precision goal of 0.1 s, will come online in the near future. It features an even larger storage volume than UCN$\tau$ and allows a combination of two complementary measurement techniques: detecting protons directly from neutron beta decays and counting remaining neutrons after certain storage times. Its different shape and design will definitely help to verify or confront the precise lifetime results from UCN$\tau$, possibly revealing new physics in the end.

Other types of new techniques have also been developed for measuring the free neutron lifetime. In particular, the recent measurement using the space-based technique [38] provides a lifetime value of $887 \pm 14(\text{stat}) + 7/ - 3(\text{sys})$ s, which is almost identical to the "beam" value, though the uncertainties are still too large.

The following sections will be organized as follows. First, we will review past neutron lifetime measurements in detail, paying particular attention to fundamental issues and extricating and clarifying nuances in different techniques. In addition, we will discuss the connections between the neutron lifetime puzzle and the unitarity problem of CKM. Next, we will introduce the newly proposed phenomenological model of $n - n'$ oscillations, including a brief history of its development and core ideas, and explain how it can explain the neutron lifetime anomaly and other immediate consequences, such as CKM non-unitarity. Lastly, we will review several experimental proposals, including new neutron lifetime measurements, that can test the unique predictions of the new mirror matter theory.

## 2. Evaluation of Neutron Lifetime Measurements

Free neutrons with a lifetime of about 15 min are known to undergo $\beta$ decays via $n \rightarrow p + e^- + \bar{\nu}_e$ due to the weak force. The two mostly applied approaches in recent decades for measuring the lifetime are the so-called "beam" and "bottle" approaches. The "beam" approach is to measure the neutron flux and decaying products such as electrons and protons from $\beta$ decays at the same time. The first better than 10 s result of $891 \pm 9$ s was reported in 1988 [22] by detecting the extracted and accelerated protons with a proportional drift counter [65]. A more promising technique using a quasi-Penning trap for confining protons was later developed resulting in an improved value of $889.1 \pm 4.8$ s [23]. Based on the same in-beam technique, the BL1 collaboration has provided so far the best "beam" result of $887.7 \pm 1.2_{\text{stat}} \pm 1.9_{\text{sys}}$ s [4,5]. An early attempt at a rather different technique, using TPC and pulsed neutron beams to detect electrons from neutron $\beta$-decays and protons from the $^3$He(n,p)t reaction simultaneously, gave a rough lifetime value of $878 \pm 31$ s [66]. Based on the same method, a better value of $898 \pm 10_{\text{stat}}\binom{+15}{-18}_{\text{sys}}$ s was recently reported for an improved experiment at J-PARC [24]. See Figure 1 and Table 1 for a compilation of all these results.

Although there are fewer "beam" measurements, they are at least consistent, unlike the "bottle" data. There are actually two different kinds of "bottle" approaches. One uses material traps that have to apply certain extrapolation techniques to remove the wall effects, in particular, due to the above-mentioned mysterious anomalous UCN losses. There have been three different extrapolation techniques applied in such experiments. The mean free path scaling (mfp scaling) method could provide a nearly model-free approach to obtain the true $\beta$-decay rate but was used only once. Another technique uses the so-called $\xi$-ratio, assuming that the UCN loss rate scales with its upscattering rate and then extrapolate the lifetime from different surface-to-volume ratios. The third one uses a model-dependent UCN loss function to deduce the normalized loss rate $\gamma$ that depends on both the UCN energy and the trap dimensions. An extrapolated lifetime can then be obtained using this $\gamma$-scaling technique. If new physics exists, each of these extrapolation methods could be affected differently.

The other "bottle" approach uses magnetic traps that, in principle, should not be susceptible to wall losses. However, the results are actually more puzzling. There have been several different techniques developed for confining neutrons under strong magnetic fields. The first of such experiments was the use of a magnetic storage ring called NESTOR at the UCN facility of the Institut Laue–Langevin (ILL) in France, which gave a lifetime value of $877 \pm 10$ s [33]. An Ioffe-type design of magnetic traps was installed at NIST as the first three-dimensional magnetic trap of UCNs [67,68]. More magnetic traps have since been built using small permanent magnets to form Halbach arrays, combined with solenoid fields to confine UCNs.

All the smaller magnetic traps, such as the Ioffe-type NIST trap [69], HOPE [34], and $\tau$SPECT [36,37], have produced very low values for neutron storage lifetime, sometimes more than 100 s lower than "accepted" lifetime values [20]. On the contrary, the much larger magnetic trap, UCN$\tau$, meanwhile produced the current most precise lifetime measurement of $877.75 \pm 0.28_{\text{stat}}\binom{+0.22}{-0.16}_{\text{sys}}$ s [9]. The irony is so remarkable that one should seriously consider what kind of new physics is at play.

Below we will evaluate these experiments and associated techniques of the two types of traps in detail and reveal their potential issues or better yet signs of new physics.

### 2.1. Ultra-Cold Neutrons in Material Traps

For the "bottle" approach with material traps, ideally one needs to find materials with a lossless perfect surface. The effective Fermi potential $V$, due to coherent nuclear scattering, is typically in the range of 100–300 neV for materials suitable for storing UCNs. For example, $V(\text{Be}) = 252$ neV and $V(\text{Cu}) = 168$ neV [70]. Total reflection for any angle of incidence on a given surface would occur if the UCN energy $E$ satisfies $E < V$.

Unfortunately, no material is lossless, even when considering no new physics, due to the effects of neutron absorption and inelastic scattering. In general, the case of UCNs bouncing off a smooth material surface can be described by quantum mechanics using a complex potential $U = V - iW$, where the Fermi potential $V$ represents the reflection barrier and $W$ accounts for neutron absorption and inelastic scattering. So the average loss probability per bounce is given by [70]

$$\bar{\mu}(E) = 2\eta \left[ \frac{V}{E} \sin^{-1}\left(\frac{E}{V}\right)^{1/2} - \left(\frac{V}{E} - 1\right)^{1/2} \right] \tag{1}$$

where the UCN loss factor $\eta = W/V$ is constant for the given material. For the case of $E \ll V$, the loss probability can be simplified as,

$$\bar{\mu}(E) = \frac{4\eta}{3}\sqrt{\frac{E}{V}} \tag{2}$$

which means that the loss probability should essentially go to zero when the UCN energy is low enough, even for an imperfect surface. Note that its low-energy behavior may be totally changed when new physics has to be considered. In particular, the new $n - n'$ oscillation model (introduced in the next section) will set a lower limit (a constant depending on the mixing strength) for $\mu(E)$ even at zero UCN energy.

The perfect materials for UCN measurements have never been found despite decades of effort. One of the best materials that we have found so far is hydrogen-free fully fluorinated polyether (PFPE) Fomblin oil or greases, which typically show a low loss probability per bounce of $\lesssim 2 - 3 \times 10^{-5}$ [71,72] that depends on the temperature.

*2.2. First Beautiful "Bottle" Measurement Using Mfp Scaling*

In the late 1980s, Mampe led a great effort to develop the so-called MAMBO (short for MAMpe BOttle) device for the first high-precision lifetime measurement. In particular, they applied a novel model-free mfp scaling technique to extrapolate their results to obtain the true $\beta$-decay lifetime. Readers are referred to the original publications [25,73] for the technical details of this beautiful experiment. Here we just give a sketch of their main ideas.

A novel technique with a volume-adjustable Fomblin-coated UCN storage vessel [71,72] was applied to measure the neutron lifetime [25]. They made an experimental design to measure the lifetime while considering the wall losses,

$$\frac{1}{\tau_n} = \frac{1}{\tau_\beta} + \bar{\mu}(v)\frac{v}{l} \tag{3}$$

where $\tau_\beta$ is the true $\beta$ decay lifetime, $l$ is the mean free path, and $v$ is the mean velocity of neutrons. One critical idea in their work is to scale the UCN storage time $t_n(i)$ with the mean free path $l(i)$ like [72]

$$\frac{t_1(i)}{t_1(j)} = \frac{t_2(i)}{t_2(j)} = \frac{t_2(i) - t_1(i)}{t_2(j) - t_1(j)} = \frac{l(i)}{l(j)} \tag{4}$$

such that neutrons of any given velocity would make the same number of collisions in different geometric configurations. More importantly, this condition ensures that the mean neutron velocity is the same, and the last term of Equation (3) depends linearly on $1/l$ for a given set of measurements with different volumes.

They did several sets of measurements corresponding to several groups of mean neutron velocities. Fig. [2] in Ref. [25] is reproduced here as Figure 2 to show the remarkable results. The linearity is as expected above. More impressively, all the extrapolations from different velocity groups converge at the same point with the condition of zero-collision or infinite mean free path ($1/l = 0$), leading to the neutron $\beta$ decay lifetime $\tau_\beta = 887.6 \pm 1.1$ s [25]. Considering other systematic uncertainties, a larger error of $\pm 3$ s

was reported in the final value [25]. Their result is in excellent agreement with a direct measurement via the "beam" approach ($887.7 \pm 1.2(\text{stat}) \pm 1.9(\text{sys})$ s) that was reported a quarter of a century later [5].

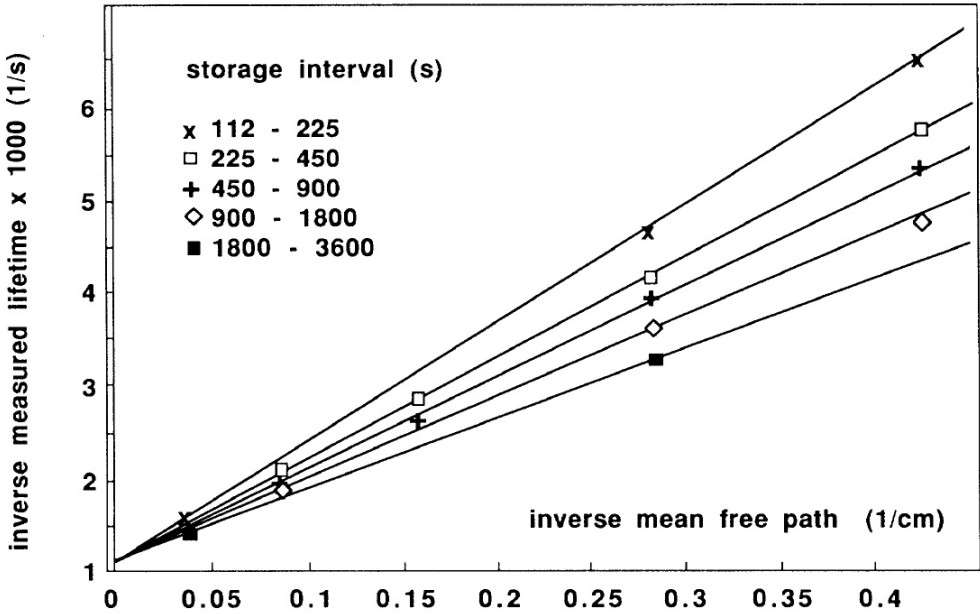

**Figure 2.** The remarkable result of the first high-precision experiment (MAMBO) [25] using the mfp scaling technique is shown. The figure is reproduced from Fig. [2] of Ref. [25].

Note that this mfp scaling extrapolation technique does not assume anything about the loss function $\bar{\mu}(v)$. It could be the same as Equation (1) or anything else. If new physics exists, it just needs to depend on the bounce rate as well, and the proposed $n - n'$ oscillations do. So this method gives the true $\beta$ decay lifetime even with new physics present, and it is no surprise that it agrees very well with later "beam" measurements. Unfortunately, the Mampe1989 experiment with the mfp scaling technique was never repeated later. The so-called MAMBO II result published in 2010 [28], though using a similar apparatus, applied the model-dependent $\gamma$-scaling technique instead. A later effort of modifying Mampe1989's result using very sophisticated quasi-elastic scattering corrections [26] is doubtful at best if not a hindsight bias to attempt to reconcile it with other "bottle" measurements.

### 2.3. Other Extrapolation Techniques in Material Trap Measurements

The $\xi$-ratio extrapolation method, first introduced in Mampe1993 [29] as shown in Table 1, assumes that the ratio of all other UCN losses (e.g., capture reaction losses) to those from inelastic scattering via wall reflections remains constant. By measuring the ratio (i.e., $\xi$-ratio) of detected inelastically scattered neutrons in two different setups with different wall surface areas, one can cancel out the wall effects and obtain the $\beta$-decay rate. However, new physics, if present, may not induce a loss rate similar to the inelastic rate. For example, the $n - n'$ oscillation effects, which we will discuss later, can be induced not only by inelastic scattering but also via incoherent elastic scattering, which does not necessarily scale with inelastic scattering.

In addition, this technique has also been plagued by additional corrections in these measurements that are large and sometimes seemingly abnormal. For example, the analysis of Mampe1993 was called into question by Ignatovich1995 [30], although it was later rebutted by Bondarenko1996 [31]. It is worth noting that Walter Mampe, who led both experiments of Mampe1989 and Mampe1993, was diagnosed with cancer in 1990 and died on 8 July 1992. As a result, he did not fully participate in the analysis and writing of Mampe1993 since it was submitted on 18 December 1992.

The $\gamma$-scaling technique originated in the early version of the Gravitrap experiment at the Petersburg Nuclear Physics Institute (PINP) with final results reported in Nesvizhevskii1992 [27]. Instead of using the mean free path as in Equation (3), this method rewrites it as $1/\tau_n = 1/\tau_\beta + \eta\gamma$ where the $\gamma$ factor includes all the energy dependence. Then this scaling technique uses Equation (1) to calculate the $\gamma$ factor for a given geometry of the trap, leading to an extrapolation to $\tau_\beta$. It is obviously model-dependent. Worse yet, Equation (1) has long been known for its problems in describing the behavior of UCNs at low energies where new physics like $n - n'$ oscillations may dominate.

Therefore, the $\gamma$-scaling method is problematic when new physics are present. In addition, the original work published in Nesvizhevskii1992 [27] was later withdrawn by one of its authors according to the PDG compilation [10]. Even with the same $\gamma$-scaling technique, the results from the two experiments by the same group using Gravitrap and Gravitrap2 (shown in Table 1), respectively, showed a moderate 2.5$\sigma$ tension. The issue may lie in the different sizes of the two traps as the Gravitrap2 device is much larger than the Gravitrap.

As discussed above, the $\xi$-ratio and $\gamma$-scaling techniques are prone to failure when dealing with new physics, unlike the mfp scaling technique. This may be the reason why the "bottle" approach has provided discrepant results over the long history of its lifetime measurements.

### 2.4. Anomalies in Magnetic Traps

One of the important milestones for neutron lifetime measurements has been the development of magnetic traps. The idea is very neat, as UCNs can be confined by strong magnetic fields of several teslas, due to a neutron's non-zero magnetic moment. In an ideal magnetic trap, there will be no collisions between UCNs and trap walls, and in principle, a perfect surface is no longer required, unlike material traps.

The first nearly perfect measurement was carried out by the UCN$\tau$ collaboration [6]. They measured the neutron lifetime to be $877.7 \pm 0.7(\text{stat}) + 0.4/ - 0.2(\text{sys})$ s, and it was soon updated to an even more precise value of $877.75 \pm 0.28(\text{stat}) + 0.22/-0.16(\text{sys})$ s [9]. This is currently the most precise lifetime measurement, and more impressively, it does not require systematic corrections larger than the quoted uncertainties. However, its measured neutron lifetime is still about 1% lower than the best "beam" result [5], further strengthening the 1% anomaly.

Unfortunately, UCN$\tau$ is currently the only magnetic trap that can produce such precise results. It has a large volume, and its bowl-shaped design is different from any other trap. As we will address later in Section 3.3, this design ensures a constant mean free flight time (i.e., independent of UCN energy) for UCNs, which may accidentally be a perfect way to hide new physics.

Other less precise magnetic traps, by contrast, have typically produced much lower, sometimes severely discrepant lifetime values. For example, the 20-pole cylindrical PNPI trap (smaller than UCN$\tau$ in size) was run at ILL and a neutron storage lifetime of $874.6 \pm 1.7$ s (lower than that of UCN$\tau$) was published [35]. At the same facility (ILL), a much smaller magnetic trap (HOPE), as shown in Figure 3, was used to measure the neutron lifetime as well [34]. The HOPE trap was designed with a very thin cylindrical volume, and a movable UCN remover rod at the top was used to measure the lifetime at two different heights of 65 and 80 cm. In another case, the remover rod was not used at all, which effectively used the full height of 1.2 m of the trap. Three different storage lifetimes were observed in these three cases: $\tau_s(\text{no remover}) = 712 \pm 19$ s, $\tau_s(80\,\text{cm}) = 824 \pm 32$ s, and $\tau_s(65\,\text{cm}) = 835 \pm 36$ s. Although large uncertainties were quoted, it seems to indicate that higher reflection rates cause more losses of UCNs, even in a magnetic trap, resulting in much lower lifetime values. Note that the HOPE experiment reported a larger final lifetime value of $887 \pm 39$ s after applying questionable large wall loss corrections, which is very uncommon for magnetic trap measurements.

Another magnetic trap τSPECT [36,37], similar to HOPE in size, but placed horizontally, confines UCNs radially with a Halbach octuple array of permanent magnets and longitudinally by the superconducting coils. It started early-phase measurements of neutron lifetime at the reactor TRIGA in Mainz, Germany. Again, similar to the results from HOPE, very low storage lifetimes in τSPECT were preliminarily reported in the Ph.D. dissertations of the group: $713 \pm 47$ s from Kahlenberg2020 [36] and $859 \pm 16$ s from Ross2021 [37].

The first three-dimensional magnetic trap of UCNs, designed in 1994 [74] and later operated at NIST [67,68], was also similar in size to both HOPE and τSPECT. However, it applied a very unique Ioffe-type design, consisting of a large magnetic quadrupole to trap UCNs radially and a pair of solenoids to trap them axially, as shown in Figure 4. The most striking feature is that the trap is loaded with isotopically pure superfluid $^4$He liquid to produce in situ UCNs using the superthermal down-scattering approach [75]. A first result of $750^{+330}_{-200}$ s was reported in 2000 [69]. Then it seemed that it could never obtain large enough values to be consistent with "accepted" lifetime values from other measurements. In 2017, an inconclusive but very disturbing value of $\tau_n = 707 \pm 20$ s was reported in an unpublished Ph.D. dissertation [20].

The lifetime results from these small magnetic traps are too low—so much so that both the HOPE and the Ioffe-type NIST trap projects were discontinued, and τSPECT has yet to publish any lifetime results. Could new physics be dependent on the neutron bounce rate or trap geometry?

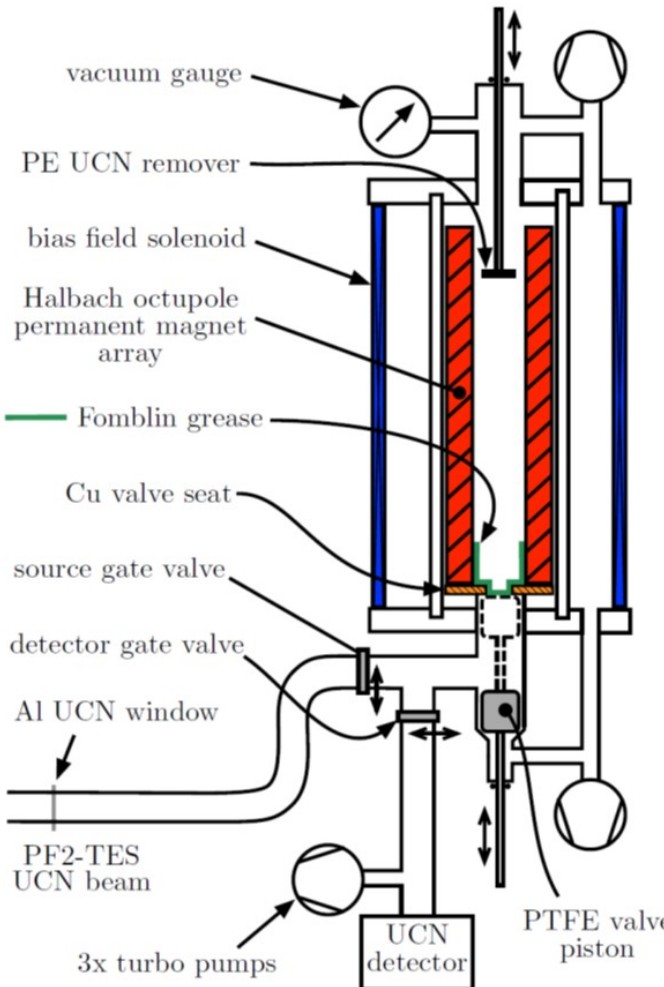

**Figure 3.** Magnetic trap HOPE at ILL. Taken from Ref. [34].

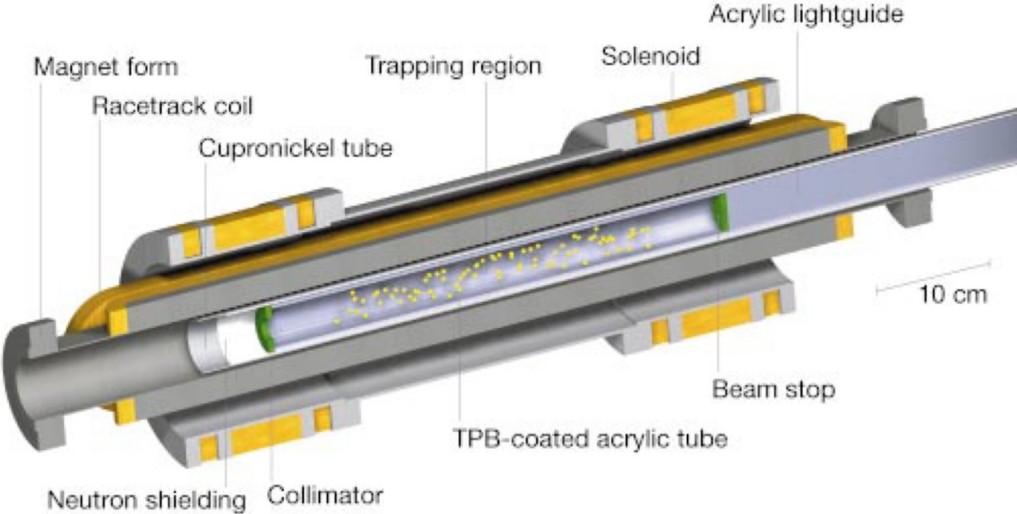

**Figure 4.** Ioffe-type magnetic trap at NIST. Taken from Ref. [69].

### 2.5. Neutron Lifetime and CKM Unitarity

Here we will evaluate the matrix elements $V_{us}$ and $V_{ud}$ of the CKM matrix by disentangling contributions from meson decays and nuclear superallowed $0^+ \rightarrow 0^+$ transitions. In particular, we consider cases in which new physics may cause more significant in-medium effects in nuclei for superallowed decays when compared to simpler systems like mesons and neutrons. Then we will compare such results with what we can infer from neutron lifetime measurements. The same approach outlined in Ref. [21] is followed below with some numerical updates.

The best direct constraint on $V_{us}$ is by measurements of the semileptonic $K_{l3}$ decays via $f_+(0)|V_{us}| = 0.21634(38)_K(3)_{HO}$ where correlation effects are considered [76,77]. The form factor at zero momentum transfer $f_+(0)$ is calculated to be $0.9698(17)$ by the lattice QCD approach [78]. The best value for the matrix element $V_{us}$ is then [77],

$$|V_{us}|(\text{meson}) = 0.22308(39)_K(39)_{latt}(3)_{HO} \qquad (5)$$

where the errors come from $K_{l3}$ decay data, lattice QCD, and high order corrections [76]. Note that a recent update by the PACS collaboration [79] presents a lower value of $f_+(0) = 0.9615(10)\binom{+47}{-3}(5)$ that leads to a higher value of $|V_{us}| = 0.2250\binom{+5}{-12}$. However, due to its very large upper systematic uncertainty ($+47$) for $f_+(0)$, its tension with the FLAGS average [78] is merely at a significance level of $< 1.6\sigma$.

The ratio of the radiative inclusive rates for $K_{\mu 2}^\pm$ and $\pi_{\mu 2}^\pm$ decays sets $f_{K^\pm}/f_{\pi^\pm}|V_{us}/V_{ud}| = 0.27600(29)_{exp}(23)_{RC}$ [76] where the FLAG averaged lattice QCD calculations give the ratio of the isospin-broken decay constants $f_{K^\pm}/f_{\pi^\pm} = 1.1932(21)$ [78]. The best value using the most recent updates is therefore [76],

$$|V_{us}/V_{ud}|(\text{meson}) = 0.23131(24)_{exp}(41)_{latt}(19)_{RC}. \qquad (6)$$

The matrix element $V_{ud}$ can then be obtained from measurements of meson decays using Equations (5) and (6),

$$|V_{ud}|(\text{meson}) = 0.9645(32). \qquad (7)$$

On the other hand, $V_{ud}$ determined from the superallowed $0^+ \rightarrow 0^+$ decays [80] using the reassessed transition-independent radiative correction $\Delta_R^V$ [81] with its current average value of $\Delta_R^V = 0.02454(18)$ [81,82],

$$|V_{ud}|(0^+ \rightarrow 0^+) = 0.97373(11)_{exp}(9)_{RC}(27)_{NS} = 0.97373(31). \qquad (8)$$

For neutron $\beta$ decays, the matrix element $V_{ud}$ can be written as,

$$
\begin{aligned}
|V_{ud}|^2 &= \frac{2\pi^3}{G_F^2 m_e^5 f_n \tau_n (1 + 3\lambda^2)(1 + \delta_R')(1 + \Delta_R^V)} \\
&= \frac{5024.46(30)\ \text{sec}}{\tau_n (1 + 3\lambda^2)(1 + \Delta_R^V)}
\end{aligned}
\tag{9}
$$

where the Fermi constant $G_F = 1.1663788(6) \times 10^{-5}\ \text{GeV}^{-2}$ [10], $m_e = 0.51099895000(15)$ MeV is the electron mass [10], the neutron-specific radiative correction $\delta_R' = 0.014902(2)$ [83], the phase space factor $f_n$ is 1.6887(1) [83,84], and natural units ($\hbar = c = 1$) are used for simplicity. The 1% difference in neutron $\beta$-decay lifetime $\tau_n$ between measurements from "beam" and "bottle" experiments leads to the discrepant $V_{ud}$ values according to Equation (9).

More recent measurements on the ratio of the axial-to-vector couplings $\lambda = g_A/g_V$ especially after 2002 have provided more reliable values [84] and its current best value of $\lambda = -1.27641(56)$ comes from the PERKEO III measurement [85]. We can also obtain $\lambda_{\text{ave}} = -1.27600(49)$ that is averaged over the measurements after 2002 using modern devices of aCORN, aSPECT, UCNA, PERKEO II, and PERKEO III [85–90] as shown in Table 2. Using the neutron $\beta$-decay lifetime of $\tau_n = 888.2 \pm 2.0$ s from the averaged "beam" values as shown in Table 1, we can obtain the matrix element,

$$
\begin{aligned}
|V_{ud}|(\text{beam } \tau_n) &= 0.9684(12) \text{ for } \lambda(\text{PERKEO III}), \\
&= 0.9686(11) \text{ for } \lambda_{\text{ave}}.
\end{aligned}
\tag{10}
$$

By taking the most precise lifetime value from UCN$\tau$2021 with a large magnetic trap [9], we can also obtain the matrix element of the "bottle" approach similarly,

$$
\begin{aligned}
|V_{ud}|(\text{bottle } \tau_n) &= 0.97414(42) \text{ for } \lambda(\text{PERKEO III}), \\
&= 0.97440(38) \text{ for } \lambda_{\text{ave}}.
\end{aligned}
\tag{11}
$$

**Table 2.** List of adopted values of the ratio of the axial-to-vector couplings $\lambda = g_A/g_V$ from experiments after 2002 using modern devices of aCORN, aSPECT, UCNA, PERKEO II, and PERKEO III [85–90] and their averaged value $\lambda_{\text{ave}}$, which is different from the recommended value of PDG2022 [10]. Note that a scale factor employed in PDG2022 makes its error much larger due to the inclusion of many discrepant old data. Here only the aSPECT result shows a tension at the $2.9\sigma$ level with the rest and hence we do not apply any additional scale factor on the error of the average value.

| Apparatus | Method | Reference | $\lambda$ |
|---|---|---|---|
| aCORN | $\beta - \bar{\nu}_e$ correlation a | Hassan2021 [90] | $-1.2796(62)$ |
| aSPECT | $\beta - \bar{\nu}_e$ correlation a | Beck2020 [89] | $-1.2677(28)$ |
| PERKEO III | $\beta$-asymmetry A | Maerkisch2019 [85] | $-1.27641(45)_{\text{stat}}(33)_{\text{sys}}$ |
| UCNA | $\beta$-asymmetry A | Brown2018 [88] | $-1.2772(20)$ |
| PERKEO II | $\beta$-asymmetry A | Mund2013 [87] | $-1.2748(8)_{\text{stat}}(^{+10}_{-11})_{\text{sys}}$ |
| PERKEO II | p-asymmetry C | Schumann2008 [86] | $-1.275(6)_{\text{stat}}(15)_{\text{sys}}$ |
| Averaged value | | | $\lambda_{\text{ave}} = -1.27600(49)$ |

In terms of CKM unitarity, the matrix element $|V_{ub}| = 0.00382(20)$ [10] is negligible. Then, we can easily check the unitarity of the CKM matrix for the first row of $|V_u|^2 = |V_{ud}|^2 + |V_{us}|^2 + |V_{ub}|^2$ using $V_{ud}$ values derived from different approaches. The results are shown in Figure 5 and Table 3. The $V_{ud}$ value determined purely from meson decays is obviously consistent with that from the "beam" n-lifetime approach while the result from nuclear superallowed $0^+ \to 0^+$ decays agrees better with that from the "bottle" method.

However, the two groups definitely contradict each other at a significance level of about $4\sigma$–$5\sigma$.

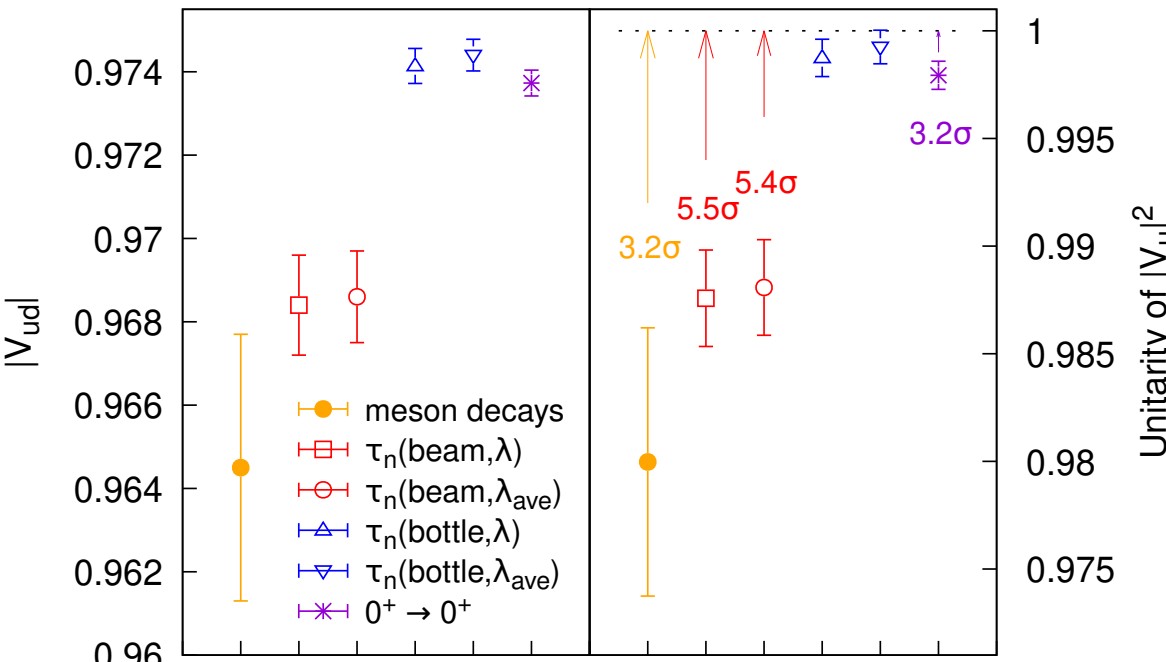

**Figure 5.** $V_{ud}$ values from different types of decay measurements and their deviations from unitarity of the CKM matrix for the first row of $|V_u|^2 = |V_{ud}|^2 + |V_{us}|^2 + |V_{ub}|^2$ are plotted.

**Table 3.** $V_{ud}$ values from different types of decay measurements are listed and their deviations in $\sigma$-level from unitarity of the CKM matrix for the first row of $|V_u|^2 = |V_{ud}|^2 + |V_{us}|^2 + |V_{ub}|^2$ are shown based on $|V_{us}| = 0.22308(55)$ from Equation (5) and $|V_{ub}| = 0.00382(20)$ [10].

| Measurement Type | $|V_{ud}|$ | $|V_u|^2$ | Deviation |
|---|---|---|---|
| meson decays | 0.9645(32) | 0.9800(62) | $3.2\sigma$ |
| neutron $\tau_n(\text{beam},\lambda)$ | 0.9684(12) | 0.9876(22) | $5.5\sigma$ |
| neutron $\tau_n(\text{beam},\lambda_{\text{ave}})$ | 0.9686(11) | 0.9881(22) | $5.4\sigma$ |
| neutron $\tau_n(\text{bottle},\lambda)$ | 0.97414(42) | 0.99874(87) | $1.5\sigma$ |
| neutron $\tau_n(\text{bottle},\lambda_{\text{ave}})$ | 0.97440(38) | 0.99925(78) | $1.0\sigma$ |
| nuclear $0^+ \to 0^+$ | 0.97373(31) | 0.99793(65) | $3.2\sigma$ |

One important conclusion we can draw from Figure 5 is that the conventional CKM matrix is most likely not unitary if the "beam" method gives the correct neutron $\beta$-decay lifetime and we will be confident that new physics exists at an about $5.5\sigma$ level if so. It is not surprising that new physics could impact the more complicated nuclear $0^+ \to 0^+$ decay systems, rather than simpler mesons, in a subtle way. Even with much larger uncertainty assigned in the latest review on $0^+ \to 0^+$ decay data [80], its $V_{ud}$ value still results in a very significant deviation from unitarity of CKM at a level of $3.2\sigma$ (compared to what used to be a level beyond $5\sigma$ [21]). As we shall present later, the newly proposed mechanism of $n - n'$ oscillations could be the best candidate to explain the non-unitarity of the CKM matrix.

### 3. Mirror Matter Theory

*3.1. A Brief History*

In 1956, Lee and Yang proposed their groundbreaking theory on parity violation in weak interactions [91], which challenged the prevailing belief in symmetric fundamental laws. The confirmation of this theory was soon conclusively demonstrated in Wu's $^{60}\text{Co}$

$\beta$-decay experiment [92]. Due to this stubborn belief in symmetry, Landau immediately proposed strict $CP$ (charge conjugation and parity) symmetry as the true and preserved symmetry between matter and antimatter [93].

However, $CP$-violation was first discovered in neutral kaon decays in 1964 [94]. This discovery shortly triggered the first proposal of mirror matter theory in 1966 by Kobzarev, Okun, and Pomeranchuk [95]. In their work, they speculated that the ordinary and mirror sectors of particles share the same gravity, but are completely decoupled in strong and electromagnetic interactions. They assumed that there is possibly some connection between the two sectors in weak interactions. The coupling of the two sectors via $K^0$ was then discussed for the search of possible mirror $K^0$ mesons [96].

In 1974, Lee connected $CP$-violation to spontaneous symmetry breaking [97], which, as we shall see, can serve similarly as a mechanism for mirror symmetry breaking.

The discovery of $CP$-violation also started speculation of possible $n - \bar{n}$ type oscillations by Kuzmin in 1970 [98]. In the late 1970s and early 1980s, similar ideas were also pursued by a host of American physicists [99–102]. However, experimental limit of $< 10^{-23}$ eV on the possible $n - \bar{n}$ mass splitting [39] make such oscillations unlikely.

After a quiet period of time in studies of the mirror matter idea, the topic was revived in the early 1980s, in particular, on possible cosmological and astrophysical manifestations [103–105]. Notably, it was demonstrated [105] that mirror matter theory could be consistent with both big bang nucleosynthesis (BBN) and the cold dark matter hypothesis in the standard cosmology model $\Lambda$CDM [106]. Considering a chaotic inflation model, Hodges presented the feasibility of mirror baryons as dark matter [107]. To be compatible with the observed $^4$He abundance produced during BBN, the temperature of the mirror sector has to be less than half of the temperature of the ordinary matter [105,107].

As it was realized that perfect mirror symmetry is not possible, various types of explicit yet feeble interactions between the two sectors were proposed. For example, the $U(1)$ photon or so-called kinetic mixing was applied to break the mirror symmetry [108]. Foot and his collaborators proposed an extension to the Standard Model (SM) with mirror particles using this scheme [109]. Based on the kinetic mixing model, they have studied a wide range of problems under the mirror matter theory [110]. In particular, Foot became one of the most enthusiastic proponents of mirror matter theory and he even published a popular science book on this topic [111] besides dozens of academic articles.

Another scheme was by introducing the 6-quark interaction term between the two sectors in the work of Ref. [43]. Its main advocate, Berezhiani, is another major figure with many publications in the recent development of mirror matter theory. However, all these early mirror models (see reviews in Refs. [43,110,112,113]) are not satisfactory due to the introduction of some ad hoc interaction mechanisms between the ordinary and mirror sectors. A historic note on the early development of mirror matter theory can be found in Ref. [114].

Mirror matter as dark matter is probably the most enticing idea and motivating factor in early works on mirror matter theory. Early studies also tried to understand neutrino oscillations with certain models of mirror matter theory [115,116]. However, it turned out that the neutrino oscillation puzzle can be explained perfectly by the generation mixing mechanism. The next advancement in mirror matter theory was to try to use it to explain the neutron lifetime anomaly [43], leading to an exciting and promising direction for this line of research. However, the understanding of the neutron lifetime issue was probably not correct. The community has shown a strong bias toward the "bottle" approach, believing that the "bottle" lifetime gives the true beta decay lifetime, which, unfortunately, misguided the development of mirror matter models.

Contrary to the common belief, it is most likely the "beam" approach that measures the true $\beta$-decay lifetime while the "bottle" method shows the neutron disappearing rate due to ordinary-mirror neutron ($n - n'$) oscillations. Based on this new understanding, a new rather exact two-parameter phenomenological $n - n'$ oscillation model was proposed to explain both the neutron lifetime anomaly and dark matter [50]. It was immediately

applied to evolution and nucleosynthesis in stars to provide a better understanding of stellar nuclear processes leading to the formation of progenitor cores of white dwarfs and neutron stars [51].

Motivated by an earlier mirror model study on ultra-high energy cosmic rays [117], the new model could present an even better understanding of such cosmic rays [52]. In particular, it predicts that the 2nd Greisen–Zatsepin–Kuzmin (GZK) cutoff due to mirror Cosmic Microwave Background (CMB) radiation should be much steeper at about $2 \times 10^{20}$ eV. It also provides a probe to determine the third or cosmological parameter of the model—the temperature ratio of mirror-to-ordinary sectors $T'/T$.

The new model also generalized the $n - n'$ oscillations into a universal mixing mechanism for neutral hadrons at the quark level but in a topological way. In particular, it predicts an unexpectedly large invisible decay branching fraction of neutral kaons. A consistent picture of the origin of both baryon asymmetry and dark matter in the early Universe was presented using kaon and neutron oscillations with new insights for the electroweak and QCD phase transitions and B-violation topological processes [53].

The new oscillation mechanism also naturally explains the unitarity problem of the CKM matrix. Most importantly, various feasible experiments are proposed to test concrete unique predictions of the new theory, including measurement of neutron lifetime anomalies in narrow magnetic traps or under super-strong magnetic fields [21], and detection of unexpectedly large branching fractions of invisible decays of long-lived neutral hadrons [58].

Based on the new phenomenological mirror matter model, an extension to the Standard Model with mirror matter was proposed to understand the mass hierarchy, nature of neutrinos, and dark energy [54]. In particular, a new understanding of mirror symmetry and supersymmetry was presented, and the energy scales of neutrino masses and dark energy were shown to be related to the tiny mass splitting scale between ordinary and mirror sectors due to staged quark condensation and spontaneous mirror symmetry breaking.

Eventually, a new theoretical framework in terms of a series of supersymmetric mirror models under different spacetime dimensions was proposed to potentially explain the arrow of time and the big bang dynamics [55,56]. Under the new framework, gravity is understood as an emergent classical phenomenon from inflated smooth spacetime, and supersymmetric mirror models provide an understanding of microphysics of Schwarzschild black holes as $2D$ boundaries of $4D$ spacetime [57,60]. A new set of first principles were then proposed as the foundations of the new framework, i.e., the quantum action principle that provides the formalism, the consistent observation principle that sets physical constraints and symmetries, and the spacetime inflation principle that determines physical contents (particle fields and interactions) [59]. Under these guiding principles, the supersymmetric mirror models can be naturally constructed to study various phases of the universe at different spacetime dimensions and the dynamics between the phases.

Most recently [61], the concept of mirror symmetry as an orientation symmetry of local spaces was further explored. Its connections to T-duality [118] and Calabi–Yau mirror symmetry [119] in string theory were established. Many developments in string theory were found to provide a solid foundation for the new framework of mirror matter theory. In particular, supersymmetric mirror models in $4D$ spacetime can be constructed from the combination of two chiral heterotic strings [120]. String theory is clearly a very powerful mathematical tool for further developing the new mirror matter theory.

We would like to clarify that other partial or seemingly related proposals in the literature are different from the above-discussed mirror matter theory. For example, the twin-Higgs model is just an incomplete mirror model for the Higgs particle only. The brane-world model seems to be an extension to mirror symmetry but is most likely just a mathematical tool lacking physical principles, similar to the relationship between string theory and supersymmetric mirror models. The many world interpretations of quantum mechanics by Everett, the multiverse vision from string theory, and any other parallel

world/universe ideas are completely different from the mirror matter theory discussed here.

More details for the general framework of mirror matter theory and its solutions to various enigmas in fundamental physics and cosmology will be reviewed in a forthcoming publication [121]. In this review, we will focus on introducing the phenomenological part of the new theory, i.e., $n - n'$ oscillations for an explanation of the neutron lifetime anomaly and the CKM unitarity puzzle, and especially for further tests of the model in the laboratory.

### 3.2. Phenomenological Model of $n - n'$ Oscillations

To be clear, theoretical aspects and considerations for new physics discussed throughout this review are based mostly on two published papers [50,53] and two unpublished ones [21,58]. Many new updates (except for Ref. [60]) on the mirror matter theory mentioned in the previous subsection have not been published yet. Therefore, the author takes sole responsibility for the views expressed here.

In the new framework of mirror matter theory, there are no cross-sector gauge interactions between particles, but a universal mixing mechanism between ordinary and mirror neutral hadrons can mediate particle oscillations between the two sectors in a topological way due to spontaneous mirror symmetry breaking [50,61]. The effect is more dramatic for longer-lived neutral hadrons such as neutrons and $K^0$. In particular, $n - n'$ oscillations become the most active messenger between the two sectors.

The main idea is that the spontaneously broken mirror symmetry causes a universal mass splitting between the two sectors of particles on a relative breaking scale of $\sim 10^{-15}$–$10^{-14}$ [50]. For neutral hadrons like neutrons, their mass eigenstates are no longer aligned with their mirror eigenstates. We can then construct the $n - n'$ oscillation model, in a way similar to the model of ordinary neutrino oscillations due to family or generation mixing [122].

The unitary mixing between ordinary and mirror neutrons can be defined as,

$$\begin{pmatrix} \psi_n \\ \psi_{n'} \end{pmatrix} = \begin{pmatrix} \cos\theta & \sin\theta \\ -\sin\theta & \cos\theta \end{pmatrix} \begin{pmatrix} \psi_{n1} \\ \psi_{n2} \end{pmatrix} \tag{12}$$

where $\psi_n$ and $\psi_{n'}$ are on the mirror basis, $\psi_{n1}$ and $\psi_{n2}$ are on the mass basis, and $\theta$ is the mixing angle.

The time evolution of $n - n'$ oscillations in the mirror representation obeys the Schrödinger equation,

$$i\frac{\partial}{\partial t}\begin{pmatrix} \psi_n \\ \psi_{n'} \end{pmatrix} = H\begin{pmatrix} \psi_n \\ \psi_{n'} \end{pmatrix} \tag{13}$$

where natural units ($\hbar = c = 1$) are used for simplicity, the Hamiltonian $H$ for oscillations in vacuum can be easily obtained as,

$$H = H_0 = \frac{\Delta_{nn'}}{2}\begin{pmatrix} -\cos 2\theta & \sin 2\theta \\ \sin 2\theta & \cos 2\theta \end{pmatrix} \tag{14}$$

and hence the probability of $n - n'$ oscillations in vacuum is [50],

$$P_{nn'}(t) = \sin^2(2\theta)\sin^2(\frac{1}{2}\Delta_{nn'}t). \tag{15}$$

Here $\sin^2(2\theta)$ denotes the mixing strength of about 0.9–2 $\times 10^{-5}$ [50,58], and $t$ is the propagation time, assumed to be much shorter than the neutron $\beta$-decay lifetime for the convenience of the following discussion. $\Delta_{nn'} = m_{n2} - m_{n1}$ is the small mass difference of the two mass eigenstates in a possible range of $10^{-6}$–$10^{-5}$ eV [53,58], leading to an intrinsic oscillation time scale of nanoseconds. Note that the equation is valid even for relativistic neutrons, and in this case, $t$ is the proper time in the particle's rest frame.

If neutrons travel in a medium, such as the dense interior of a star or strong magnetic fields in a laboratory, the Mikheyev-Smirnov-Wolfenstein (MSW) matter effect (first proposed and observed for solar neutrinos) [123,124] may be important. That is, coherent forward scattering with other nuclei, or interactions between the neutron's magnetic moment and the magnetic field, can affect the oscillations by introducing an effective interaction term in Hamiltonian,

$$H_I = \begin{pmatrix} V_{\text{eff}} & 0 \\ 0 & 0 \end{pmatrix}. \tag{16}$$

where the effective potential can be obtained as

$$V_{\text{eff}} = \begin{cases} \dfrac{2\pi}{m_n} \sum_i b_i n_i & \text{for dense matter} \\ -\vec{\mu}_N \cdot \vec{B} & \text{for magnetic fields} \end{cases} \tag{17}$$

where $\mu_N \simeq -6 \times 10^{-8}$ eV/T is neutron's magnetic moment, $m_n$ is the neutron mass, $n_i$ is the number density of nuclei of $i$-th species in the medium, and $b_i$ is the corresponding bound coherent scattering length. Therefore, the modified in-medium Hamiltonian can be written as,

$$H = H_M = \frac{\Delta_{nn'}}{2} \begin{pmatrix} -\cos 2\theta + V_{\text{eff}}/\Delta_{nn'} & \sin 2\theta \\ \sin 2\theta & \cos 2\theta - V_{\text{eff}}/\Delta_{nn'} \end{pmatrix} \tag{18}$$

and the corresponding transition probability for a time-independent potential $V_{\text{eff}}$ is

$$P_M(t) = \sin^2(2\theta_M) \sin^2\left(\frac{1}{2}\Delta_M t\right) \tag{19}$$

where $\Delta_M = C\Delta_{nn'}$, $\sin 2\theta_M = \sin 2\theta / C$, and the matter effect factor is defined as,

$$C = \sqrt{(\cos 2\theta - V_{\text{eff}}/\Delta_{nn'})^2 + \sin^2(2\theta)}. \tag{20}$$

One important consequence of the medium effect is that $n - n'$ oscillations can become resonant, similar to the case of solar neutrino flavor oscillations [124]. The resonance condition is $\cos 2\theta = V_{\text{eff}}/\Delta_{nn'}$, meaning that the effective potential $V_{\text{eff}}$ is nearly equal to the $n - n'$ mass difference since $\cos 2\theta \sim 1$ for $n - n'$ oscillations. This condition depends on the unknown sign of the mass difference $\Delta_{nn'}$, which is most likely positive due to constraints of stellar revolution [51]. When it resonates, the effective mixing strength is nearly one compared to the vacuum value of the order of $10^{-5}$. However, these resonant conditions are not easy to realize. For example, the matter density has to be in the range of about $10^2$–$10^3$ g/cm$^3$, which is only possible in the dense interior of a star, or super-strong magnetic fields around 50–200 T are required.

On the other hand, incoherent collisions or interactions of a neutron in the medium can reset the neutron's oscillating wave function or collapse it into a mirror eigenstate. For material or magnetic traps of UCNs, the coherent medium effect is negligible and the mean free flight time $\tau_f$ of a neutron between incoherent bounces is on the order of 0.1–1 s is much longer than the vacuum oscillation scale of $\sim 10^{-9}$ s. Therefore, by averaging out the propagation factor in Equation (15), we obtain the transition rate due to $n - n'$ oscillations in a trap,

$$\lambda_{nn'}(\text{bottle}) = \frac{1}{2\tau_f} \sin^2(2\theta) \tag{21}$$

which depends simply on the mixing strength constant $\sin^2(2\theta)$ and the mean free flight time $\tau_f$ that depends on neutron energy spectrum and trap geometry. It contributes to the apparently measured lifetime as follows,

$$\frac{1}{\tau_n} = \frac{1}{\tau_\beta} + \lambda_{nn'}. \tag{22}$$

Considering the decay of a free neutron or in general a neutral hadron with a mean lifetime of $\tau$, we can restore the factor of $\exp(-t/\tau)$ in the oscillation probability in Equation (15). Then we can obtain the branching fraction of ordinary-mirror oscillations or so-called invisible decays as follows,

$$B_{\text{inv}} = \frac{1}{2}\sin^2(2\theta)\frac{(\Delta\tau)^2}{1 + (\Delta\tau)^2}. \tag{23}$$

For the "beam" approach, neutrons interact only once via either in-flight $\beta$ decay or capture reaction in the detector at the end of the flight path. Therefore, their loss probability due to $n - n'$ oscillations is much smaller,

$$P_{nn'}(\text{beam}) = \frac{1}{2}\sin^2(2\theta) \sim 10^{-5} \tag{24}$$

Note that the two parameters in the model, the mixing strength $\sin^2(2\theta)$ and the mass difference $\Delta$, can be determined experimentally. However, they are not independent of each other. Constrained by the consistent picture of explaining the origin of both dark matter and baryon asymmetry of the universe from $n - n'$ and $K^0 - K^{0'}$ oscillations [53], an elegant relation between the two parameters is shown in Figure 6. It is remarkably consistent with current experimental constraints from neutron lifetime measurements. In addition, a third cosmological parameter, $T'/T$, defining the temperature ratio of the two worlds, has to be fixed by observation as well. We will discuss how we can more accurately measure the first two parameters of the model in laboratory experiments in the following section.

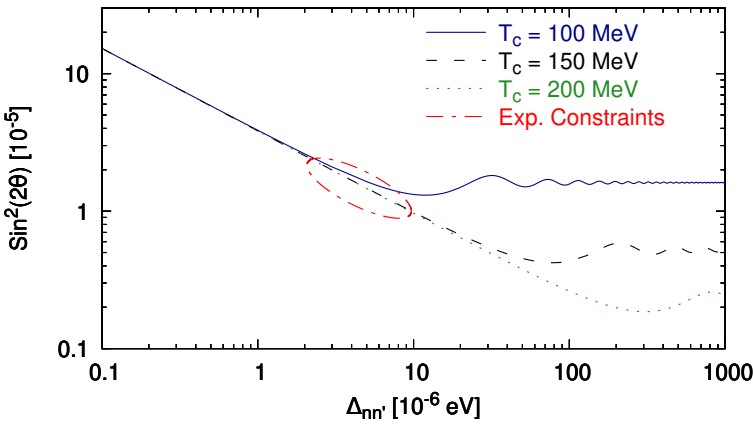

**Figure 6.** The mixing strength $\sin^2(2\theta)$ is shown as a function of the mass difference $\Delta_{nn'}$, assuming a mirror-to-ordinary baryon ratio of 5.4 in the universe. The QCD phase transition temperature is varied to be $T_c = 100, 150, 200$ MeV, respectively. Adapted from Ref. [53].

### 3.3. Explanation of Neutron Lifetime Anomaly and CKM Unitarity Puzzle

Now we can use the new $n - n'$ oscillation mechanism to explain the puzzles in neutron lifetime and unitarity of the CKM matrix.

From Equation (24), we see that the $n - n'$ oscillation effect is about two orders of magnitude lower than the precision ($\sim 10^{-3}$) of the "beam" approach. That is why the

true $\beta$-decay rate is measured in the "beam" approach. On the other hand, UCNs typically bounce $\sim 10^3$ times within one mean lifetime in a trap, resulting in more losses via $n - n'$ oscillations. Therefore, a typical 1% discrepancy from the $\beta$-decay lifetime can be obtained from Equation (21).

A new simulation [125] using this $n - n'$ oscillation model for the previous experiment with the Ioffe-type NIST trap has remarkably reproduced the observed lifetime anomaly of $707 \pm 20$ s [20]. The UCN$\tau$ experiment is also consistent with the new model based on a recent simulation result for the mean free flight time of $0.33 \pm 0.08$ s [126]. Furthermore, these two experiments can be used to constrain the mixing strength $\sin^2(2\theta)$ within a range of $0.8$–$2 \times 10^{-5}$ [58].

Note that an early estimate of the lower limit of $\sin^2(2\theta)$ [50] has essentially presented the same result. If UCNs are prepared at a much higher temperature than their kinetic energies, the corresponding Maxwell–Boltzmann velocity distribution will be proportional to $v^2$ in this extreme case. Therefore the mean velocity of UCNs can be calculated from $\bar{v} = \frac{3}{4} v_{\max}$ in the extreme case or $\bar{v} < \frac{3}{4} v_{\max}$ in more realistic cases when more energetic neutrons will oscillate more frequently, resulting in a softer energy spectrum. From the trap dimensions of UCN$\tau$, the UCN mean free path of $l \sim 75$ cm can be estimated. The maximum UCN velocity $v_{\max} \sim 309$ cm/s can be inferred from the trap potential of 50 neV. Therefore, we can estimate the mean free flight time $\tau_f = l/\bar{v} \geq 0.34$ s and hence a lower limit for the mixing strength $\sin^2(2\theta) \geq 8 \times 10^{-6}$.

In a compiled list of the UCN loss coefficient $\eta$ for PFPE Fomblin oil as a function of temperature [127], it is shown that $\eta$ starts to level off and eventually settle down to around $2$–$6 \times 10^{-6}$ when the temperature drops below about 180 K. This could be a sign of oscillation effects dominating at lower temperatures, resulting in the above-mentioned anomalous wall loss effect. If we take the average value of $\eta = 4 \times 10^{-6}$, then roughly we can obtain $\sin^2(2\theta) \sim 4\eta = 1.6 \times 10^{-5}$, which agrees well with other constraints [58].

Because of the way how the new $n - n'$ oscillation effect depends on the bouncing rate in a trap, the mfp scaling technique can effectively remove it from its extrapolation, which explains why the MAMBO result from Mampe1989 [25] agrees very well with the "beam" lifetime values. On the other hand, the other two extrapolation techniques, especially the $\gamma$-scaling method, are poor in removing the $n - n'$ oscillation effect, and as a result, they tend to give rise to smaller lifetime values.

Although using the same $\gamma$-scaling technique, the results from the two experiments using Gravitrap and Gravitrap2 differ by 3 s or $2.5\sigma$ as shown in Table 1. Such tension can be explained via $n - n'$ oscillations since Gravitrap2 as an upgrade is much larger than Gravitrap, leading to a much lower collision rate between UCNs and the trap walls.

The results from the magnetic HOPE trap, though quoted with large uncertainties, seem to indicate that the measured lifetime values depend dramatically on the trap height. Again, this can be explained under the new model. For such a narrow cylindrical trap, as shown in Figure 3, the mean free path of a trapped UCN is nearly constant, i.e., determined by the trap diameter. In contrast, the average energy of UCNs is higher when the trap lid or the remover rod is placed at a higher position, resulting in a larger bounce rate or a shorter mean free flight time. Therefore, the $n - n'$ oscillation effect becomes more significant, leading to a shorter lifetime for a taller trap. This also explains why all the small magnetic traps such as HOPE, $\tau$SPECT, and the Ioffe-type NIST trap, give rise to much smaller lifetime values.

As for the UCN$\tau$ trap, its unique bowl-shaped design for the magnetic Halbach array ensured its high-precision results. Assuming that the trap shape is parabolic, one can estimate that for a given height $h$, its surface area is $S \sim h^{3/2}$, and its volume is $V \sim h^2$. The mean free path can then be estimated from $l = 4V/S \sim \sqrt{h}$. Meanwhile, the mean speed of UCNs can be estimated as $v \sim \sqrt{E} \sim \sqrt{h}$ due to gravitational confinement from the top. In the end, we obtain the mean free flight time nearly energy-independent as $\tau_f = l/v \sim$ constant. Such a first-order estimate indicates that the $n - n'$ oscillation effect is constant

for UCN$\tau$ and their results should be independent of the height for filling UCNs, leading to a consistently lower value than the $\beta$-decay lifetime.

Another striking feature of the new $n - n'$ oscillation model is that it predicts that the conventional CKM matrix is not unitary. Indeed, using the "beam" lifetime, we can obtain, for the first row of the CKM matrix, $|V_u|^2 = 0.9876(22)$ as shown in Table 3, which is about 5.5$\sigma$ away from unitarity. The missing part is due to the topological mirror oscillation process [61]. To maintain overall unitarity, we can infer an effective element $|V_{uu'}| \simeq 0.11$ [21,58]. Note that $V_{uu'}$ does not manifest in the same way as the other perturbative matrix elements due to gauge charge conservation, since ordinary and mirror quarks have separate gauge interactions.

Using the information from the oscillation strengths of $n - n'$ and $K^0 - K^{0'}$, we can also calculate other effective topological elements of $|V_{dd'}| \simeq 0.063$ and $|V_{ss'}| \simeq 0.018$ [21,58]. Such oscillation elements can lead to unexpectedly large invisible decay branching fractions for relatively long-lived neutral hadrons such as $K^0_L$, $K^0_S$, $\Lambda^0$, and $\Xi^0$ [58]. Any experimental confirmation of such oscillations or invisible decays would further verify the non-unitarity of the CKM matrix.

## 4. Further Laboratory Tests

The mechanism of universal neutral ordinary-mirror hadron oscillations predicted in this mirror matter theory could be used to test unexpectedly large invisible decay rates of long-lived light neutral hadrons. In particular, large yet realistic estimates of invisible decay branching fractions from $K^0_L$, $K^0_S$, $\Lambda^0$, and $\Xi^0$ due to such oscillations are calculated to be $9.9 \times 10^{-6}$, $1.8 \times 10^{-6}$, $4.4 \times 10^{-7}$, and $3.6 \times 10^{-8}$, respectively [58]. These significant invisible decays are either readily detectable at existing accelerator facilities or at least testable in the near future. A recent search for invisible decays of the $\Lambda$ baryon motivated by the new mirror model has just been published [128], although they have not yet reached the sensitivity required to detect them. Surprisingly, no experimental upper limits exist for $K^0$ invisible decays. Fortunately, the long-planned NA64 experiment at CERN will be conducted in the near future with very promising experimental sensitivities of $Br(K^0_L \to \text{invisible}) \lesssim 10^{-6}$ and $Br(K^0_S \to \text{invisible}) \lesssim 10^{-8}$ [129]. However, we will focus on more feasible laboratory tests of $n - n'$ oscillations below.

### 4.1. Magnetic UCN Traps with Different/Narrow Geometries

The new mirror model predicts that the neutron loss probability, by oscillating into its mirror counterpart, is about $\frac{1}{2}\sin^2(2\theta) \sim 10^{-5}$ for each incoherent bounce of a neutron against trap walls, or effectively each 180-degree change of direction in a magnetic trap. Therefore, narrower magnetic traps or traps with smaller mean free paths, in other words, the higher bounce rate for UCNs, can more significantly amplify the effects of $n - n'$ oscillations, resulting in smaller lifetime values (e.g., easily $> 100$ s less than the $\beta$-decay lifetime). Such traps can provide the most convincing tests for the new mirror matter model.

For example, the Ioffe-type NIST trap as shown in Figure 4 would be the best option. In particular, the pure $^4$He superfluid filled in the trap serves as the detection medium, and at the same time produces UCNs in situ via the superthermal down-scattering process. This is ideal as liquid $^4$He does not absorb ultra-cold neutrons and does not incoherently scatter them either. Possible improvements could include lining the trap with a scintillator detector to monitor marginally trapped UCNs and developing a technique of pulse-shape discrimination to separate $\beta$-decay events from those due to $^3$He contamination. The early strong evidence of such large anomalies like the $707 \pm 20$ s value from the NIST trap [20] should motivate the community to restart such measurements in the near future.

The HOPE project or a similar effort is also worth restarting. The vertical cylindrical design of the HOPE trap, shown in Figure 3, is well suited for testing the geometrical dependence of the mirror oscillation effects in a systematic way. By varying the trap height, different lifetime values can be expected according to the mirror oscillation model. For such

smaller narrow traps, the effect is so large that high precision is not necessarily required to discover new physics.

Meanwhile, high-precision measurements with different large magnetic traps will also be helpful. Currently, UCN$\tau$ is the only experiment of its kind, and its design, as discussed above, seems to be perfectly hiding the $n - n'$ oscillation effect. Fortunately, PENeLOPE [64], another large magnetic trap, will come online in the near future. Its different geometry will ensure a different result from that of UCN$\tau$ should the world have a mirror sector.

On the other hand, the true $\beta$-decay lifetime also needs to be determined more accurately, in particular, using the "beam" approach. The BL2 [62], an immediate upgrade of BL1 currently deployed at NIST, is expected to achieve a better uncertainty of $\lesssim$1 s. The future much larger BL3 device aims to reduce the uncertainty to an even better value of $\lesssim$0.3 s [63]. By comparing the difference between the $\beta$-decay lifetime and magnetic trap results, one can not only verify the new theory convincingly but also accurately determine one of the two parameters of the $n - n'$ oscillation model—the mixing strength $\sin^2(2\theta)$.

*4.2. Resonant $n - n'$ Oscillations in Super-Strong Magnetic Fields*

The phenomenon of resonant oscillations due to the medium effect, when the effective potential is close to the mass difference between ordinary and mirror neutrons, is another unique prediction from the mirror matter model. Although it is not possible for us to test such resonant $n - n'$ oscillations in super-dense matter of about $10^2$–$10^3$ g/cm$^3$ in a laboratory, it is feasible to test this phenomenon under super-strong magnetic fields in a laboratory. According to Equation (20), most likely resonant oscillations require magnetic fields to be around 50–200 T. A recent publication [49] has shown null results for testing such oscillations under much weaker magnetic fields up to a few teslas, which essentially ruled out a previous mirror matter model [48]. Nonetheless, such results are consistent with the new model discussed here and encourage us to test it under much stronger fields.

It is not trivial to produce such high magnetic fields in a laboratory. Direct-current high fields up to 45.5 T have recently been demonstrated in a very compact magnet setup with new conductor material and a novel design [130]. Such continuous DC fields would have been ideal for the test as one can gradually increase the field until it reaches the resonant value where 50% of neutrons with proper alignment with the field will oscillate into mirror neutrons. Unfortunately, we do not have strong enough DC fields yet.

However, super-strong pulsed fields are readily available at several facilities around the world, e.g., at the Pulsed Field Facility of National High Magnetic Field Laboratory (NHMFL-PFF) at LANL [131]. In particular, NHMFL-PFF has several 65 tesla and one 100 tesla non-destructive pulsed magnets. The pulse duration is roughly on the order of milliseconds, which is much longer than the $n - n'$ oscillation time scale of about a nanosecond.

Should the actual resonant field exceed 100 T, there is another option available: the 300 tesla single-turn magnet at NHMFL-PFF. Its pulse duration of about 5 microseconds is much shorter but still longer than the $n - n'$ oscillation scale.

For pulsed magnets, we need to solve the Schrödinger equation as in Equation (13) with the time-dependent effective potential $V_{\text{eff}}(t)$. On the mirror basis, we can obtain the evolution of the wavefunction when a neutron goes through a pulsed magnet as follows,

$$\begin{aligned}
\frac{\partial^2 \psi_n(t)}{\partial t^2} + \left( \frac{\Delta_{nn'}^2}{4} \left( \left( \cos 2\theta - \frac{V_{\text{eff}}(t)}{\Delta_{nn'}} \right)^2 + \sin^2 2\theta \right) + \frac{i}{2} \frac{\partial V_{\text{eff}}(t)}{\partial t} \right) \psi_n(t) &= 0 \\
\frac{\partial^2 \psi_{n'}(t)}{\partial t^2} + \left( \frac{\Delta_{nn'}^2}{4} \left( \left( \cos 2\theta - \frac{V_{\text{eff}}(t)}{\Delta_{nn'}} \right)^2 + \sin^2 2\theta \right) - \frac{i}{2} \frac{\partial V_{\text{eff}}(t)}{\partial t} \right) \psi_{n'}(t) &= 0
\end{aligned} \tag{25}$$

where $V_{\text{eff}}$ can be replaced with the time-dependent magnetic field $B(t)$ according to Equation (17), and the initial condition is that the evolution starts from a pure ordinary neuron wavefunction,

$$\begin{pmatrix} \psi_n(t=0) \\ \psi_{n'}(t=0) \end{pmatrix} = \begin{pmatrix} 1 \\ 0 \end{pmatrix}. \tag{26}$$

Equation (25) can then be numerically solved using the so-called Numerov's method, and more details on such studies will be discussed in a forthcoming paper [132].

At or near the resonant field (i.e., $\mu B \sim \Delta_{nn'}$), the oscillation effect is too large to be missed. If successfully detected, it will provide the best measurement of the $n - n'$ mass splitting $\Delta_{nn'}$—one of the two key parameters of the new theory.

The sign of $\Delta_{nn'}$ is still unclear although the application of this model to stellar evolution [51] indicates that it is probably positive. Future experiments using polarized neutron beams under such super-strong magnetic fields may help determine the sign.

*4.3. Other Possible Tests*

Based on the new model, the neutron disappearing rate via $n - n'$ oscillations can be significant for neutrons scattered in a solid/liquid medium with low absorption but high incoherent cross sections. Deuterium is indeed such an element with a very low absorption cross-section of $5.19 \times 10^{-4}$ b and much higher incoherent scattering cross-section of 2.05 b for thermal neutrons [133]. Using the $n - n'$ per-bounce loss rate of $10^{-5}$, we can estimate the invisible fraction of thermal neutron losses in deuterium to be about 4%, which may be detectable, possibly in a heavy water ($D_2O$) detector.

The $n - n'$ oscillation effects could also manifest in the quasi-free medium of a halo nucleus. One example is $^{11}$Be, a one-neutron halo nucleus. It has a 13.76 s $\beta$-decay half-life with a strong $\beta$-delayed particle decay ($\beta\alpha$) branch of 3.3% [134]. A rare $\beta p$ decay branch of $^{11}$Be $\rightarrow$ $^{10}$Be was measured using the accelerator mass spectrometry technique [135,136] and the TPC technique [137]. The results are not consistent, though they favor an unexpectedly high branching ratio of $10^{-6}$–$10^{-5}$ that can not be explained by theoretical calculations unless there exists an unexpected near-threshold resonance [138,139]. If such an anomaly persists, one possible explanation would be that the $n - n'$ oscillation effect may help overcome the barrier of penetrability.

## 5. Conclusions and Outlook

The puzzles in modern neutron lifetime studies since the late 1980s are reviewed and the compelling evidence of new physics is discussed. One of the most important insights from the analysis of the lifetime anomaly is that the "beam" approach most likely gives the true $\beta$-decay lifetime, while new physics causes anomalous lifetimes in "bottle" measurements. To summarize new physics, the neutron lifetime measurements suggest strong evidence for $n - n'$ oscillations under the newly proposed mirror matter model, which is presented in detail. The current experimental constraints on the only two parameters of this rather exact model are given as the mixing strength $\sin^2(2\theta)$ in between $8 \times 10^{-6}$ and $2 \times 10^{-5}$ and the $n - n'$ mass splitting $\Delta_{nn'}$ within 3–11 $\times 10^{-6}$ eV.

Reviving small or narrow magnetic traps, such as the Ioffe-type NIST trap and the HOPE trap, should be given priority as the new model predicts surprisingly large anomalies (>100 s) in lifetime measurements using such traps. High-precision large magnetic traps, particularly those with a shape different from UCN$\tau$'s, such as PENeLOPE, should also help clarify the situation by revealing the geometric dependence predicted by the new model. Additionally, better magnetic trap experiments will provide better measurements of the $n - n'$ mixing strength of the new model.

Another experimental program that should also receive immediate support is to search resonant $n - n'$ oscillation effects under super-strong magnetic fields of about $10^2$ T. Pulsed magnets at NHMFL-PFF, though mainly focused on material science, can provide high enough fields for such measurements. Successful results from this program can not only

help verify another unique prediction by the new model but also provide a very accurate measurement of the $n - n'$ mass splitting parameter.

Should the existence of the mirror sector of the universe be confirmed in proposed experiments, immediate and enormous consequences will follow. We would know that the CKM matrix is not unitary, that dark matter is made of mirror matter, etc. A very rich and new research direction beyond what we have learned from the Standard Model may be on the horizon.

**Funding:** This work is supported in part by the faculty research support program at the University of Notre Dame.

**Data Availability Statement:** No new data were created or analyzed in this study. Data sharing is not applicable to this article.

**Acknowledgments:** The author would like to thank Benjamin Grinstein for his invitation and encouragement during the journey of writing and publishing this review article.

**Conflicts of Interest:** The author declares no conflict of interest.

## Notes

1    re-analysis of Mampe1989, see text for details.
2    withdrawn by A. Serebrov according to PDG [10].
3    re-analysis of Mampe1993 [29], but rebutted by Bondarenko1996 [31].

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
