# Peer review of "Neutron Lifetime Anomaly and Mirror Matter Theory"

_universe, doi:10.3390/universe9040180_

Round 1

Reviewer 1 Report

I have reviewed the manuscript "Neutron lifetime anomaly and mirror matter theory" by W. Tan. This paper discusses the beam-bottle neutron lifetime discrepancy in the context of possible new physics explanations. I did not find anythiing really new here; all of the ideas presented can already be found in the published literature on this. Nevertheless this paper does serve as a useful and interesting review. I have two substantial comments that should be addressed and corrected prior to publication.

1. In Table I and elsewhere in the paper, the author presents a number of "neutron storage time" results as neutron lifetime measurements.  Such results were not intended to be, and should not be, regarded as measurements of the neutron beta decay lifetime. For example, in the NIST Ioffe trap experiment He-3 contamination in the superfluid helium cell contributed to neutron losses, so the storage times were lower than the beta decay lifetime. Similary the HOPE "lifetime results" in Table I were reported as storage times in Ref. 34; the lifetime result is clearly presented as 887 ± 39 s, the result that should appear in Table I. Also the tauSPECT "results" presented are not lifetime results. It is misleading to include neutron storage times in Table I. Those should be removed, and this distinction should be made more clear throughout the paper.

2. On p6 the author states "the most feasible solution could be the newly proposed n − n′ oscillation model 141 [21, 50], which regards the “beam” lifetime as the true β-decay lifetime". This is not necessarily true. In one popular hypothesis, the presence of a magnetic field oscillation resonance may cause the neutron flux in the proton trap of the NIST beam lifetime apparatus to be lower than that on the neutron counter, causing the beam lifetime result to be systematically too high. While it is reasonable to suggest that n − n′ oscillation may be contributing to the neutron lifetime discrepancy, the outcome depends on details of physics that is not yet known, so the author should not presume at this point what the "true" beta decay lifetime is.

After these corrections are made throughout the paper I believe this paper will be suitable for publication in Universe.

Reviewer 2 Report

In this paper, the author reviewed in detail the current experimental status related to the neutron lifetime anomaly and its connection to the CKM matrix unitarity, and proposed a solution based on the n--n' oscillation picture. The author suggested that the "beam" experiments provide the correct measurement for the beta decay lifetime, in contrast to the "mainstream" belief in the community that the "bottle" measurements are correct. The author also proposed several experiments to test the theory, for example to study the dependence of "bottle" neutron lifetime on the bottle size, and to look for n--n' oscillations in super-strong magnetic fields.

Even if disregarding the proposed mirror matter theory, I think this is a very comprehensive review that provides readers many useful information about the neutron lifetime measurement. I may recommend its publication after the author takes into account several issues as follows:

1. One reason the author used as a support for the "beam" value of Vud is that it is in agreement with the "meson" determination of Vud, namely the combination of Vus from semileptonic kaon decays and Vus/Vud from leptonic kaon/pion decays. But this conclusion depends on the value of Vus from Kl3, which in turn depends sensitively on the lattice calculation of f+(0); the latter is not without dispute. For example, a recent series of Nf=2+1 calculations of f+(0) by the PACS collaboration returned f+(0)=0.9615, about 0.9% smaller than the FLAG average (see Phys.Rev.D106,094501 and references therein). If using this value, then the resultant Vud will move towards the neutron (bottle) and superallowed result. The author should at least mention this possibility.

2. Given that the setup of superallowed nuclear beta decay lifetime experiments resembles the "beam" experiments for neutron, one would naturally expect Vud(beam) to be closer to Vud(superallowed) if the author's theory was correct. But the reality is the opposite: Vud(bottle), instead, is closer to Vud(superallowed). I think this requires some explanations. 

3. It appears to me that only neutral hadrons will oscillate to their mirror partners in this picture, and that would lead to some observable consequences. For instance, this will predict that the value of Vus determined from semileptonic K^+ decay is different from that obtained from semileptonic K^0 decay. What would be the required experimental precision level in order to observe such a deviation?

4. In Table II the author deduced an average value of lambda, but they did not include the scale factor S which PDG adopts to account for possible discrepancies between different experiments (which, in this case, refers to the difference between the aSPECT result and others). As a consequence, the quoted uncertainty of lambda by the author is much smaller than that in PDG. I think this needs to be justified.

5. The theory predicts that the measured neutron lifetime from "bottle" experiments depends on the bottle size. Given that there are already a number of bottle experiments, for example those in Table I, is it possible to make a plot of lifetime vs size to see if such a trend can already be observed?

6. Does the author's theory receive any constraint from the neutron mirror experiment, i.e. Ref.[49]?

Reviewer 3 Report

The main argument against the publication of this paper is that the original results and conclusion are based on results of other yet unpublished paper of the author itself. Some of them are still unpublished after more than 3 years. 

What is the cause of such delay? Is the reviewing process still on going?

Round 2

Reviewer 1 Report

I believe it is the author who is confused here. The "storage time" or "storage lifetime" which are the same thing in my usage, refers specifically to the observed exponential decay time of the population of trapped neutrons, including the effects of beta decay and other systematic losses which may or may not be well understood. In some cases the storage lifetime may be corrected for some effects but not others. The "neutron lifetime" refers specifically to the beta decay lifetime only. It is inappropriate and misleading to present these together in Figure 1 and Table 1, as it confuses acknowledged systematic effects with more interesting differences that may be due to new physics.  It is also very misleading to imply that the result in Craig Huffer's PhD thesis was meant as a measurement of the beta decay lifetime - it was well acknowledged by Craig and his collaborators that unaccounted systematic losses were likely present - it should be interpreted as a "corrected storage lifetime",  corrected for known systematics only.  As the author did not apparently take this objection from my original report seriously, I must at this stage recommend rejection of this revised manuscript.

Reviewer 2 Report

The author has adequately responded to my concerns, so I recommend the publication of this manuscript. Nevertheless, I would still like to emphasize that, the fact that the author is unsure about the impact of their theory on Vud from superallowed nuclear decays represents a possible caveat. I strongly suggest the author to perform an in-depth study of this issue, maybe in a future work. Such a study could produce a high impact to the precision physics community working on first-row CKM unitarity.